# Trastuzumab-Modified Gold Nanoparticles Labeled with ^211^At as a Prospective Tool for Local Treatment of HER2-Positive Breast Cancer

**DOI:** 10.3390/nano9040632

**Published:** 2019-04-18

**Authors:** Łucja Dziawer, Agnieszka Majkowska-Pilip, Damian Gaweł, Marlena Godlewska, Marek Pruszyński, Jerzy Jastrzębski, Bogdan Wąs, Aleksander Bilewicz

**Affiliations:** 1Centre of Radiochemistry and Nuclear Chemistry, Institute of Nuclear Chemistry and Technology, Dorodna 16, 03-195 Warsaw, Poland; l.janiszewska@ichtj.waw.pl (Ł.D.); m.pruszynski@ichtj.waw.pl (M.P.); a.bilewicz@ichtj.waw.pl (A.B.); 2Department of Biochemistry and Molecular Biology, Centre of Postgraduate Medical Education, Marymoncka 99/103, 01-813 Warsaw, Poland; damian.gawel@cmkp.edu.pl (D.G.); marlena.godlewska@cmkp.edu.pl (M.G.); 3Heavy Ion Laboratory, University of Warsaw, Pasteura 5A, 02-093 Warsaw, Poland; jastj@slcj.uw.edu.pl; 4Institute of Nuclear Physics, Polish Academy of Sciences, Radzikowskiego 152, 31-342 Cracow, Poland; Bogdan.Was@ifj.edu.pl

**Keywords:** α-nanobrachytherapy, monoclonal antibody, SKOV-3 ovarian cell line, HER2 receptor, α-emitter ^211^At

## Abstract

Highly localized radiotherapy with radionuclides is a commonly used treatment modality for patients with unresectable solid tumors. Herein, we propose a novel α-nanobrachytherapy approach for selective therapy of human epidermal growth factor receptor 2 (HER2)-positive breast cancer. This uses local intratumoral injection of 5-nm-diameter gold nanoparticles (AuNPs) labeled with an α-emitter (^211^At), modified with polyethylene glycol (PEG) chains and attached to HER2-specific monoclonal antibody (trastuzumab). The size, shape, morphology, and zeta potential of the 5 nm synthesized AuNPs were characterized by TEM (Transmission Electron Microscopy) and DLS (Dynamic Light Scattering) techniques. The gold nanoparticle surface was modified by PEG and subsequently used for antibody immobilization. Utilizing the high affinity of gold for heavy halogens, the bioconjugate was labelled with ^211^At obtained by α irradiation of the bismuth target. The labeling yield of ^211^At was greater than 99%. ^211^At bioconjugates were stable in human serum. Additionally, *in vitro* biological studies indicated that ^211^At-AuNP-PEG-trastuzumab exhibited higher affinity and cytotoxicity towards the HER2-overexpressing human ovarian SKOV-3 cell line than unmodified nanoparticles. Confocal and dark field microscopy studies revealed that ^211^At-AuNP-PEG-trastuzumab was effectively internalized and deposited near the nucleus. These findings show promising potential for the ^211^At-AuNP-PEG-trastuzumab radiobioconjugate as a perspective therapeutic agent in the treatment of unresectable solid cancers expressing HER2 receptors.

## 1. Introduction

Highly localized radionuclide therapy with radioisotopes has become a standard treatment option for many malignant tumors. Two approaches for delivering radiation to tumor sites are currently in clinical practice: systemic radionuclide therapy using targeted radiopharmaceuticals and brachytherapy with application of sealed radioactive sources. Brachytherapy is a method of radiotherapy where a closed radiation source is placed inside or near the area needing treatment. This method is effective and commonly used for curing of solid tumors, such as those involving the prostate, breast, liver, and skin [1]. Brachytherapy usually utilizes sources of 50–100 μm in size, such as ^125^I [2], ^106^Ru [3], ^192^Ir [4], ^131^Cs [5,6], ^137^Cs [7,8], or ^103^Pd [9,10] radioactive seeds, and ^90^Y immobilized in resin or glass microspheres [11,12,13]. In the case of breast cancer, brachytherapy has found application as part of a post-surgery partial irradiation strategy for patients undergoing lumpectomy [14]. The limited affinity of micro sources for tumor vasculature coupled with a significantly smaller (50–100 μm) size relative to tumor vasculature porosity (150–300 nm) causes substantial leakage of radioactivity away from the tumor site [15]. Such problems reduce the efficacy and tumoricidal activity, and increase the acute toxicity of brachytherapy agents [15]. With the fast development of nanoscience and nanotechnology, it has become attractive to prepare injectable nanoscale brachytherapy seeds [16,17], as much smaller needles can be used for injection of nanoseed colloidal solution and can reduce the trauma caused by surgical implantation [18]. In addition, it can provide an easy way to change the radiation dose by the injection of different amounts of the nanoparticles (NPs) solution [19,20]. A number of *in vitro* and *in vivo* tests on the use of liposomes with encapsulated ^186^Re have been presented [21,22,23]. These demonstrated that intraoperative use of radionuclides incorporated into liposomes could play an important role in the treatment of positive surgical margins in advanced squamous cell carcinoma of the head and neck. Despite these advances, the use of nanostructured systems must be conducted with care, given that liposomes present a high potential for release of their contents during administration [24]. Therefore, new nanosystems with higher stability have been proposed. Soares and co-workers suggested the use of silica nanoparticles functionalized with diethylenetriaminepentaacetic acid (DTPA) and labeled with the β^−^emitter ^159^Gd [25]. The obtained nanostructures remained stable over several hours in biological fluids without a significant release of their contents. Other nanoseeds, such as ^103^PdAuNPs [18], ^198^AuNPs stabilized with arabic gum [26], ^166^Ho in poly-l-lactide polymer NPs [27], and ^142^Pr_2_O_3_NPs [28], have also been prepared as potential radiopharmaceutical agents for intratumoral radiotherapy. Furthermore, in traditional brachytherapy, heterogeneous dose distribution occurs within the tumor. Recently, Reilly et al. proposed a novel targeted nanomedicine brachytherapy approach for the treatment of locally advanced breast cancer. They used intratumoral injection of gold nanoparticles (diameter: 30 nm) modified with a polyethylene glycol (PEG) polymer with an attached DOTA (1,4,7,10-tetraazacyclododecane-1,4,7,10-tetraacetic acid) chelator that complexes with the β^−^ radionuclide ^177^Lu. The conjugate links to panitumumab, which exhibits affinity to epidermal growth factor receptor (EGFR)-positive tumor cells, or to trastuzumab, which binds specifically to human epidermal growth factor receptor 2 (HER2) receptors [29,30]. Their studies on biodistribution show that Au-trastuzumab injected intratumorally is retained (~30% ID per g) with very low uptake by the organs, such as the liver and spleen [31]. They also found that targeting HER2 facilitated binding of trastuzumab and internalization in HER2-positive tumor cells in comparison to nontargeted AuNP-^177^Lu. In immunocompromised female NOD/SCID mice with HER2-overexpressing human breast cancer xenografts, treatment with trastuzumab-AuNP-^177^Lu for 16 days resulted in significant inhibition of tumor growth compared with AuNP-^177^Lu-exposed or untreated mice. In our work to increase the efficiency and selectivity of therapy, we applied a similar approach but used the much more radiotoxic α-emitter, ^211^At, instead of the β^−^ emitter, ^177^Lu. The range of emitted α-particles in tissues is only 5–10 cell diameters, which limits the deposition of a therapeutic dose to the targeted cell and its surroundings [32]. In comparison to the β^−^ particles, α-particles provide higher relative biological effectiveness, destroying more cells with a lower radiation dose. The high linear energy transfer (LET) of α-particles results in many more DNA double-strand breaks than β^−^ particles [33]. In addition, the therapeutic effect of α-particles is independent on the hypoxia state of the cells. This is an advantage over β^−^ emitters for radioimmunotherapy, as the latter depend on the creation of superoxide free radicals [34]. Among the hundreds of α-emitters, only a few α-particle-emitting radionuclides have properties suitable for developing therapeutic radiopharmaceuticals: generator-obtained ^212^Bi (*t*_1/2_ = 60 min), ^213^Bi (*t*_1/2_ = 46 min), ^226^Th (*t*_1/2_ = 30 min), ^225^Ac (*t*_1/2_ = 10 d), ^227^Th (*t*_1/2_ = 18.7 d), and the cyclotron-produced ^211^At (*t*_1/2_ = 7.2 h). Unfortunately, current supplies of medically useful α-emitters remain limited due to products isolation in nuclear weapon processing and power plant development within the United States and former Soviet Union, with the parental stock being slated for disposal [35]. Therefore, cyclotron-produced ^211^At seems to be the most promising candidate for targeted α-radiotherapy because its half-life of 7.2 h assures sufficient time for its transportation, isolation from the target, labeling process, quality control, and medical application without problems caused by long-lived daughters of α-emitting radionuclide, which is observed in the case of radionuclides (^223^Ra and ^225^Ac) that are mother isotopes for other alpha emitters (^211^Pb, ^213^Bi) [36]. A variety of ^211^At-labeled biomolecules have been examined in preclinical models, and two have reached the stage of initial clinical trials [34,37]. Unfortunately, initial results have not been very positive due to instability of the astatine–biomolecule bond in biological fluids. Although astatine is a member of the halogen group, it also shows some metallic character in the +1 oxidation state [38]. The energy carbon–halogen bond for astatine is significantly lower than for iodine, which excludes the use of elaborated iodination methods for labeling biomolecules with ^211^At [39]. On the other hand, it is well-known that iodines chemisorb on the gold metal surface and the affinity of halides on the gold surface increases in the order of F^−^ < Cl^−^ < Br^−^ < I^−^. Therefore, the F^−^ ion adsorbs on the metal surface very weakly, but the Cl^−^, Br^−^, and I^−^ anions are able to adsorb on the gold surface, forming an Au−X chemical bond [40]. This phenomenon of strong I^-^ adsorption on the gold surface was used to introduce a new method of iodination of biomolecules using a gold nanoparticle as a metal linker between ^125^I and a biomolecule [41], with ^125^I-AuNP used as a source of Auger electrons in targeted therapy [42]. Considering the increasing bond strength between the gold surface and the halogen atom in the order of F^−^ < Cl^−^ < Br^−^ < I^−^, we assumed that astatide, the heaviest halogen anion, would bind to the gold surface more strongly than iodide. As our recent theoretical [43] and experimental studies [44] have shown, astatine is very strongly chemisorbed on the gold surface and allows for gold nanoparticles to function as a linker between ^211^At and a biomolecule. The synthesized ^211^At-AuNP-substance P radiobioconjugate was very stable in human serum as well as in cerebrospinal fluid and exhibited affinity to neurokinin 1 (NK1) receptors on glioma tumor cells. Herein, we propose for the first time a novel α-nanobrachytherapy approach for selective treatment of HER2-positive breast cancer. This uses local intratumoral injection of 5-nm-diameter gold nanoparticles labeled with ^211^At and modified with PEG chains linked to trastuzumab, which bind the radiobioconjugate to HER2-positive tumor cells.

## 2. Materials and Methods

### 2.1. Materials

The chemical reagents used were as follows: gold (III) chloride trihydrate (AuCl_3_·3H_2_O), trisodium citrate dihydrate (Na_3_C_6_H_5_O_7_), PEG carboxylic acid disulfide (COOH-PEG-SS-PEG-COOH), triethylamine (99% purity), Iodogen (1,3,4,6-tetrachloro-3R,6R-diphenylglycouril), paraformaldehyde (PFA), bovine serum albumin (BSA), Triton X-100, Tween 20, goat anti-human IgG (γ-chain specific), fluorescein isothiocyanate (FITC)-conjugate (cat. no. F1641), and 4′,6-diamino-2-phenylindole dihydrochloride (DAPI). All reagents were purchased from Sigma-Aldrich (St. Louis, MO, USA). Nitric acid (65%), hydrochloric acid (35%), and methanol (99.9%) were bought in POCH (Gliwice, Poland). Anhydrous dimethylformamide (DMF) and dichloromethane (DCM) of ACS grade were obtained from BDH Prolabo VWR (Gdańsk, Polska). Trastuzumab was isolated from the commercial drug Herceptin (Roche Pharmaceuticals, Basel, Switzerland). Fluorescence mounting medium was purchased from Dako (Carpinteria, CA, USA). Serum aliquots were prepared from blood samples drawn from healthy volunteers and kept at −20 °C. The volunteers were told about the research protocol and informed consent was obtained. The study was carried out in accordance with institutional regulations. The following materials were used for cell experiments: McCoy’s medium and fetal calf serum from Biological Industries (Beth Haemek, Israel), phosphate-buffered saline (PBS), dimethylsulfoxide (DMSO), and the CellTiter 96^®^ AQ_ueous_ One Solution Reagent (MTS compound) from Promega (Mannheim, Germany). SKOV-3 cells were obtained from the American Type Tissue Culture Collection (ATCC, Rockville, MD, USA) and cultured according to the ATCC protocol.

### 2.2. Radionuclides

Astatine-211 was obtained by irradiation of natural bismuth in a ^209^Bi(α,2n)^211^At reaction, in the α beam of the U-200 cyclotron at the Heavy Ion Laboratory at Warsaw University (Warsaw, Poland). Separation of ^211^At was carried out by dry distillation under nitrogen (at a flow rate of 120 cm^3^ min^−1^). Firstly, the apparatus was flushed with nitrogen for 15 min, then the active bismuth target was placed in a quartz tube inside the furnace and heated. After 30 min, the temperature reached 650 °C for ^211^At release and was stably maintained for the next 15 min. Finally, the ^211^At activity was collected in a cold polyether ether ketone (PEEK) tube submersed in ethanol and its temperature was cooled down with liquid nitrogen to about −50/−55 °C. Due to better availability of ^131^I than ^211^At, and their similar chemical properties resulting from the neighborhood in the halogen group [45], we used the ^131^I radionuclide instead of ^211^At in several presented experiments. No-carrier-added [^131^I] Na with the specific activity of approximately 550 GBq mg^−1^ was obtained from the POLATOM Radioisotope Centre (Świerk, Poland).

### 2.3. Characterization Techniques for Nanoparticles

Size, shape, and morphology of the obtained nanoparticles and nanoparticle conjugates were characterized by transmission electron microscopy (TEM, Libra 120, Carl Zeiss NTS GmbH, Oberkochen, Germany) [46]. The hydrodynamic diameter and zeta potential (ζ) were determined by dynamic light scattering (DLS, Malvern, UK) [47]. The radioactive samples were measured in a gamma spectrometer containing a germanium detector (Canberra, Meriden, CT, USA) with a resolution of 0.8 at 5.9 keV, 1.0 at 123 keV, and 1.9 at 1332 keV. Radioactivity of ^211^At was quantified by its 89.69 keV γ-ray. The oxidation state of astatine ions was determined by paper electrophoresis (Sigma–Aldrich horizontal electrophoresis) on Whatman GF83 Glass Paper, at the gradient of 12 V cm^−1^ for 20 min and with the use of an electrolyte (0.01 M NaNO_3_). The solutions were prepared with deionized water from a Hydrolab water purification system (Hydrolab, Straszyn, Poland). All TLC (Thin Layer Chromatography) analyses were carried out on Alufolien sheets (RP-18 Merck 7.5 cm) with the use of MeOH (methanol) as a mobile phase. The radioactivity distribution on the TLC sheets or the paper electrophoresis was determined by Storage Phosphor System Cyclone Plus (Perkin-Elmer Life and Analytical Sciences, Shelton, CT, USA). Confocal imaging was performed using an LSM800 confocal microscope equipped with a plan-apochromat 63×/1.4 oil DIC M27 lens (Zeiss, Jena, Germany). Transmitted light images in the bright field were acquired using a transmitted light detector with a photomultiplier tube (T-PMT, Zeiss, Jena, Germany).

### 2.4. Synthesis of 5 nm AuNPs

AuNPs (5 nm) were synthesized in aqueous solution by chemical reduction as described previously [48]. More specifically, a mixture of sodium citrate (4.48 g, 0.877 wt%) and tannic acid (1.73 g, 1 wt%) was added to a boiling aqueous solution of chloroauric acid (93.80 g of 1.84 × 10^−2^ wt%). After the addition of the mixture to the solution, the suspension changed color from yellow to dark red. Next, the reaction mixture was stirred for an additional 15 min and was finally cooled down to ambient temperature. The concentration of AuNPs was determined as follows: the mass of one AuNp was calculated assuming the sphericity of 5 nm AuNPs. Subsequently, knowing the mass of gold used for synthesis, the numbers of AuNPs in 1 mL solution were determined.

### 2.5. Synthesis of AuNP-S-PEG-Trastuzumab Bioconjugate

AuNPs of 5 nm diameter (0.1 µg in 1 mL) were first coated with the PEG linker comprising the disulfide bridge with carboxyl groups at each end (HOOC-PEG-SS-PEG-COOH) in molar ratios of 100:1 of PEG to gold nanoparticles. After 24 h, a mixture of 2-fold excess molar amounts of 1-ethyl-3-(3-dimethylaminopropyl) carbodiimide hydrochloride (EDC) and N-hydroxysulfosuccinimide (NHS) was added to the solution of pegylated gold nanoparticles to form an amide bond with the amine group of lysine. After an additional 4 h (without purification of the excess of EDC/NHS reagents), trastuzumab (100 µg per mL of modified gold nanoparticle solution) was attached to the AuNP-PEG-NHS. The bioconjugate solution was alkalinized to pH 9.0 and the sample was stirred for 24 h at ambient temperature in an inert gas atmosphere. After this time, the product was purified in dialysis cassettes and labeled with astatine-211 according to the previous procedure.

### 2.6. Determination of the Amount of Trastuzumab-PEG-OPSS (Orthopyridyl Disulfide) per AuNP

To estimate the average number of trastuzumab-PEG molecules linked to each AuNP, radioiodinated trastuzumab-PEG-OPSS was added into the AuNP conjugation reaction. For this purpose, trastuzumab (2 mg, 10 mg mL^−1^) was labeled with ^131^I (32–37 MBq) using the Iodogen method [49]. Briefly, 10 µg of Iodogen was mixed with 100 ug of trastuzumab, 0.05 M PBS (phosphate buffer saline), and 10 MBq of ^131^I. The sample was stirred for 10 min at room temperature and then purified on a PD-10 column containing Sephadex G-25 resin. After the reaction of 2 × 10^11^ AuNPs with ^131^I-labeled trastuzumab-PEG-OPSS, the sample was loaded into the dialysis membrane (30 kDa molecular weight (MW) cutoff) and placed in water overnight to remove the excess of unbound ^131^I-labeled trastuzumab-PEG-OPSS. As the centrifugation of 5 nm size AuNPs is not possible, due to their small size, we applied the dialysis method with the use of a membrane of 30 kDa instead of Vivaspins. The next day, 1 mL of the dialysate solution was analyzed by gamma spectroscopy to measure the amount of ^131^I-labeled trastuzumab-PEG-OPSS that leeched out. The binding efficiency of ^131^I-trastuzumab-PEG to AuNPs was assessed by measuring the proportion of radioactivity coupled to AuNPs. Finally, to calculate the number of ^133^I-trastuzumab-PEG molecules attached to each AuNP, the moles of ^131^I-trastuzumab-PEG were divided by the moles of AuNP.

### 2.7. Labeling of AuNP-S-PEG-Trastuzumab Bioconjugate with ^211^At and ^131^I

The labeling of AuNP-S-PEG-trastuzumab with ^211^At and ^131^I was assessed by the adsorption of these radionuclides on gold nanoparticle conjugates. Firstly, the astatine in Na_2_SO_3_/methanol solution (200–400 µL) eluted from PEEK was evaporated and then dissolved in water. To perform the labeling, 20 μL of ^211^At^−^ activity (5–10 MBq) was added to 200 μL of AuNP-S-PEG-trastuzumab bioconjugate solution placed in Eppendorf tubes. The pH of the reaction mixture was adjusted to 6.0 by dropwise addition of HNO_3_ or NaOH and stirred for 1 h at room temperature. Labeling with ^131^I was carried out according to the same procedure as for ^211^At using 3 MBq of Na^131^I in 0.01 M NaOH. To determine the labeling efficiency, the TLC method with MeOH as mobile phase was employed [44]. In this technique, free ^211^At^−^/^131^I^−^ moves with the solvent front while labeled bioconjugate stays at the bottom of the TLC strip. The location of radioactivity on the TLC strips was measured by a Storage Phosphor System Cyclone Plus (Perkin-Elmer Life and Analytical Sciences, Shelton, CT, USA) and analyzed using Optiquant software (version 5.0, Waltham, MA, USA) supplied by the manufacturer.

### 2.8. Stability Studies of ^211^At/^131^I-AuNP-S-PEG-Trastuzumab Radiobioconjugates

^211^At-AuNP-S-PEG-trastuzumab or ^131^I-AuNP-S-PEG-trastuzumab solution (20 µL) was added to 200 µL of human serum and incubated at room temperature. Stability studies were performed after 2, 4, and 24 h on three parallel samples using the TLC method. Serum with radionuclide solutions used as a blank tests were carried out to determine the percentage of eventual binding of free ^211^At to proteins present in each solution.

### 2.9. Confocal Imaging

SKOV-3 cells (10^5^ per well) were grown in six-well plates covered with sterile glass coverslips (ϕ12 mm/#1.5; Thermo Scientific, San Jose, CA, USA). After 20 h of incubation, the medium was removed and fresh culture medium (2 mL) containing: AuNPs (3.97 × 10^12^ particles/well); or trastuzumab (210.4 µg/well); or AuNP-S-PEG-trastuzumab bioconjugate (3.97 × 10^12^ particles/well) was added. Cells grown in complete medium were used as controls. After 24 h of incubation, the coverslips were transferred to 24-well plates and cells were washed, permeabilized, and blocked as described in [50]. For trastuzumab or bioconjugate detection, samples were treated (for 1 h) with the goat anti-human IgG (γ-chain specific) fluorescein isothiocyanate (FITC)-conjugate (1:200; cat. no. F1641, Sigma-Aldrich) in blocking solution. Next, the cells were washed and stained with 4′,6-diamino-2-phenylindole dihydrochloride (DAPI; Sigma-Aldrich). Finally, coverslips were mounted using mounting medium (Dako, Carpinteria, CA, USA). A Zeiss LSM800 (with plan-apochromat 63×/1.4 oil DIC M27 lens) was used for confocal imaging (Zeiss, Jena, Germany). Fluorescence images for FITC-labeled trastuzumab and DAPI were gained at 488 and 408 nm, respectively. Transmitted light images in the bright field for Au particles or bioconjugate were acquired using a transmitted light detector (T-PMT). Images were recorded and adjusted using ZEN 2.1 software (Zeiss) and saved as TIFF files.

### 2.10. Cell-Binding Studies

To determine the affinity of ^131^I-AuNP-S-PEG-trastuzumab to HER2 receptors overexpressed on SKOV-3 cells and for comparison with the reference compound (^131^I-trastuzumab), saturation cell-binding assays were performed. Briefly, 6.0 × 10^5^ SKOV-3 cells per well were seeded into six-well plates and cultured at 37 °C overnight. After 24 h, the medium was replaced and different amounts of radiobioconjugates (varied from 0.3 to 81 nm for ^131^I-AuNP-S-PEG-trastuzumab and from 0.8 to 100 nM for ^131^I-trastuzumab) were added to the cells and incubated at 4 °C for 1.5 h. Cellular uptake was stopped by removing the supernatant; afterwards, the cells were washed twice with 1 mL of cold PBS. Finally, the cells were lysed three times with 1 M NaOH. Fractions of radiobioconjugate bound and unbound to receptors were collected and measured on a Perkin Elmer γ-counter. Nonspecific binding was assessed by co-incubation with 100-fold excess of non-radiolabeled trastuzumab (blocking experiment). Specific binding was expressed as the difference of total and nonspecific binding. Binding affinity (*K*_d_) and receptor density (*B*_max_) was estimated using the Scatchard analysis method [51]. Results are expressed as the mean ± standard deviation (SD) of three individual experiments.

### 2.11. In Vitro Cytotoxicity Assay

Cytotoxicity studies were performed using ^211^At-AuNP-trastuzumab and ^211^At-AuNPs radioconjugates. SKOV-3 cells were plated at a density of 2.0 × 10^3^ cells per well. The next day, the cells were washed twice with cold PBS and radiocompounds (0.0005–12.5 MBq mL^−1^) were added, in triplicate experiments, to a final volume of 100 µL per well. Cells were incubated at 37 °C for 24 h, washed twice with PBS to remove the unbound fraction, and subsequently incubated at 37 °C for another 48 h as described previously [52]. For analysis, the CellTiter-96^®^ AQueous-Non-Radioactive (Promega) MTS assay was used. The absorbance in wells was measured at 490 nm using the microplate reader, Apollo 11LB913 (Berthold, Bad Wildbad, Germany). Results are expressed as a percentage of cell viability relative to cells grown in medium only (control).

### 2.12. Statistical Analysis

At least three independent experiments in six replicate wells were performed for each toxicity point. Differences between radioactive samples and control were assessed using GraphPad Prism 5.1 software (GraphPad Software Inc., San Diego, CA, USA). Differences were considered statistically significant when the *p*-value was <0.05.

## 3. Results and Discussion

HER2 is overexpressed in various types of cancer cells, such as breast, lung (non-small cell), stomach, colon, and ovary [53]. About 20–30% of breast cancers overexpress HER2 [54], with expression correlating with higher tumor aggressiveness and risk of distant metastasis to the central nervous system [55]. The ability of α-particle-emitting ^225^Ac and ^211^At radionuclides to treat HER2-expressing human breast cancer cells has been assessed *in vitro* and *in vivo* [56,57]. It was found that the relative biological efficacy of ^211^At-labeled trastuzumab was about 10-fold higher than that of external beam irradiation, with a significant reduction in the survival of cancer cells achieved only by using several ^211^At atoms per cell. Further, a considerable prolongation in median survival time, including some long-term survivors, was observed [53]. Large protein molecules, such as antibodies, are not good to use together with relatively short-lived radionuclides, such as ^211^At, because their large size hinders homogeneous delivery. Additionally, intravenous administration results in slow clearance from normal tissue. Therefore, the ^211^At–trastuzumab radiobioconjugate may be more effective for local injection. The promising results obtained with intratumorally injected ^177^Lu-labeled gold nanoparticles conjugated with trastuzumab led us to use ^211^At-Au-trastuzumab as an α-particle source to kill cancer cells overexpressing HER2 receptors [30].

Our group discovered that the high affinity of gold metal to astatine allowed for the creation of radioactive nanoseeds that were easy to obtain. Gold nanoparticles with a median diameter of 5 nm were prepared using sodium citrate and tannic acid as stabilizing agents. The obtained AuNPs were stable in solution because the AuNPs adsorbed on the citrate layer provided the force of an electrostatic repulsion from the double electric layer [58]. A schematic illustration of the AuNP conjugation to trastuzumab is shown in Figure 1. At the first step, AuNPs were coated with a PEG linker, comprising a disulfide bridge with carboxyl groups at either end, using the well-recognized strong gold–sulfur bond. Mangeney et al. [59] showed that hydrophilic polymers end-capped with a disulfide can be efficiently grafted onto citrate-stabilized AuNPs in aqueous solution. The ratio of linker molecules to AuNPs was chosen so that about 50% of the surface remained covered by citrates and was available for ^211^At adsorption. In the second step, AuNP-PEG-COOH nanoparticles were conjugated to trastuzumab by amide bond formation with the amine group of lysine.

To determine the average number of trastuzumab molecules attached to one AuNP, ^131^I-labeled trastuzumab was incorporated into the reaction mixture with AuNPs. Using this method, it was established that four trastuzumab molecules were conjugated with one particle of AuNP when 7.94 × 10^13^ AuNPs were reacted with 81.04 μg of antibody. The calculation was carried out based on the mass of ^131^I-trastuzumab (81.04 µg) under the sphericity assumption of nanoparticles with a median diameter of 5 nm, as determined by TEM, and that the density of gold was 19.28 g cm^−3^.

Physicochemical properties of AuNP-S-PEG-trastuzumab bioconjugates, such as size and morphology, were characterized by the DLS and TEM techniques. The results are presented in Table 1 and Figure 2, respectively.

TEM images of AuNP-S-PEG-trastuzumab indicated that these nanoparticles were nearly spherical with a core equal to 5 nm. Analysis of the mean diameter and size distribution of AuNP-S-PEG-trastuzumab showed that the gold core of the AuNP-S-PEG-trastuzumab bioconjugate was similar to that of the citrate-coated particles. These results indicated that AuNP integrity was unaffected by PEG-trastuzumab grafting. The presence of trastuzumab molecules on the AuNP surface was confirmed by DLS measurement of the hydrodynamic diameters and zeta potentials of AuNPs, AuNP-S-PEG-COOH, and AuNP-S-PEG-trastuzumab (Table 1). The AuNP size (11.7 ± 0.3 nm), determined by DLS, was notably larger than that measured by TEM. The increase in hydrodynamic diameter after the addition of PEG, peptide, and protein was observed after modification of nanoparticles with biomolecules, as was the case for AuNPs modified by trastuzumab or folic acid elsewhere [60,61]. The difference between the zeta potential value of citrate AuNPs and AuNP-S-PEG-trastuzumab additionally confirms the surface modification. The zeta potential of AuNP-PEG-trastuzumab was −36.9 ± 0.9 mV, which provides important information on the stability of AuNP-S-PEG-trastuzumab dispersion. It reveals that the particles repel each other and that the particles do not tend to aggregate, which was confirmed by observing changes in hydrodynamic diameter within 7 days.

The synthesized AuNP-S-PEG-trastuzumab bioconjugates were labeled with ^131^I and ^211^At by adsorption radionuclides on the gold surface. The radioactivity of ^211^At ranged from 100 to 150 MBq upon arrival at the laboratory. Separation of ^211^At was performed using dry-distillation of the target material, so that the recovery yield equaled 50% of initial ^211^At activity. This routinely took around 40 min of preparation time. Aliquots of the obtained astatine were evaporated, dissolved in water, and directly used for labeling of AuNP-PEG-trastuzumab bioconjugates. The presence of the anionic form of astatine was confirmed by paper electrophoresis through comparison with the behavior of the iodide anion. The ^131^I in the iodide form was dissolved in 0.01 M NaOH.

The labeling yields of AuNPs and AuNP-S-trastuzumab with ^211^At are shown in Table 2. The AuNPs labeling efficiency with ^211^At was >99% and was significantly higher than that of another commonly used prosthetic group, *N*-succinimidyl-3-(tri-n-butylstannyl) benzoate, where the labeling efficiency ranged from 60 to 70% [33]. For the boron precursor [62] and for the Rh [1 6aneS4-diol] complex elaborated in our laboratory, the yields were equal to 80–89% and 80% [63], respectively. As presented in Table 2, ^211^At absorption on the gold surface in AuNP-S-PEG-trastuzumab conjugates was nearly the same as for naked AuNPs, indicating that the attachment of PEG-trastuzumab molecules to the AuNP surface changes the adsorption properties of AuNPs minimally. This is expected because, according to our calculations, half of the surface of gold nanoparticles is not occupied by trastuzumab molecules and ^211^At can be attached there.

Stability of ^211^At-AuNP-S-PEG-trastuzumab in human serum was tested at various time points (Table 3). The percentage of liberated ^211^At was determined using TLC analysis with methanol (MeOH) as a developing solvent. As previously examined, the intact astatinated radiobioconjugates remained at the bottom of the TLC strip (*R*_f_ = 0), while free ^211^At, released from the radiobioconjugates, moved with the solvent front (*R*_f_ = 0.9) [44].

Radiobioconjugates ^211^At-AuNP-S-PEG-trastuzumab and ^131^I-AuNP-S-PEG-trastuzumab exhibited good stability in human serum. Nearly no release of ^211^At was observed over 24 h, corresponding to more than three half-lives. As shown in Table 3, astatinated radiobioconjugates are more stable than iodinated analogues. This confirmed our hypothesis that astatide, as the biggest halogen anion in the group, formed the strongest bond with the gold surface atoms.

As ^131^I is much more readily available than ^211^At, and as iodine and astatine exhibit similar properties, we used ^131^I radionuclide instead of ^211^At in cell affinity experiments. Increasing concentrations of ^131^I-AuNP-S-PEG-trastuzumab (from 0.3 to 81 nM) and ^131^I-trastuzumab (from 0.8 to 100 nM) were incubated with SKOV-3 cells overexpressing HER2 receptors in the presence or absence of a 100-fold molar excess of trastuzumab molecules. Total, nonspecific, and specific binding affinity curves are shown in Figure 3.

The receptor affinity of the ^131^I-AuNP-S-PEG-trastuzumab radiobioconjugate was compared with the commonly known properties of ^131^I-trastuzumab (reference compound). The dissociation constant values (*K*_d_) obtained from a radioligand binding assay and the maximum number of binding sites (*B*_max_) were: *K*_d_ = 16.6 ± 4.1 nM (corresponding to 2.7 µg mL^−1^ of AuNPs), *B*_max_ = 2.89 nM for ^131^I-AuNP-S-PEG-trastuzumab, and *K*_d_ = 10.4 ± 3.2 nM and *B*_max_ = 1.75 nM for radiolabeled trastuzumab.

The *K*_d_ value obtained in our work was more than two times greater than that determined by Cai et al. for ^177^Lu-DOTA-AuNPs [29]. This discrepancy may be associated with the use of different nanoparticle sizes (Cai et al. used 30 nm AuNPs) and numbers of trastuzumab molecules attached to AuNPs. Considering that our bioconjugate contained only four trastuzumab molecules and the conjugate used by Cai et al. contained 13 trastuzumab molecules, the obtained results are consistent. A comparison of the *K*_d_ value for ^131^I-AuNP-S-PEG-trastuzumab with the *K*_d_ value for ^131^I-trastuzumab (both determined under the same conditions) indicates that attachment of 5 nm AuNPs reduces the receptor-specific affinity of a trastuzumab molecule to a small degree.

To assess the *in vitro* capability of human ovarian cancer-derived SKOV-3 cells to internalize AuNP-S-PEG-trastuzumab bioconjugate, the cells were exposed to free-AuNPs or AuNP-S-PEG-trastuzumab bioconjugate for 24 h. To determine trastuzumab’s native binding properties, the set of cells were also treated with free trastuzumab solution. Evaluation was performed using confocal imaging and is summarized in Figure 4. It was hypothesized that, under the applied conditions, free AuNPs (3.97 × 10^12^ particles per well) would sporadically enter cells, while those tied in a bioconjugate would be efficiently delivered into cells due to the high binding affinity of trastuzumab to HER2 receptors, which are overexpressed on SKOV-3 cells. The acquired images confirmed that only the AuNP-S-PEG-trastuzumab complex was effectively internalized by SKOV-3 cells. The dark spots observed on the light background (Figure 4; panel b/4) reflect AuNPs particles, while the red fluorescence signals mark trastuzumab binding (Figure 4; panel c/4). Additionally, staining the cells with DAPI, which selectively counterstains nuclei (blue fluorescence), allowed for the precise intracellular localization of the bioconjugate particles. Merged signals (Figure 4; panel d/4) revealed that bioconjugate particles successfully penetrated SKOV-3 cells and were preferentially localized in the nuclear envelope area. Nonetheless, some of the merged signals were also observed inside the nucleus, indicating that the tested bioconjugate may pass through the nuclear membrane (Figure 4; panel d/4). As expected, free AuNPs were not detected in the cells (Figure 4; panel b/3). SKOV-3 cells that were untreated (Figure 4; column 1) or treated only with trastuzumab antibody (Figure 4; column 2) served as negative and positive controls, respectively. It was demonstrated that SKOV-3 cells have high binding activity and internalize free trastuzmab (Figure 4; panel c/2).

In the next step, the influence of ^211^At-AuNPs and ^211^At-AuNP-PEG-trastuzumab radioconjugates on SKOV-3 cell viability was investigated. Cells were exposed to different activities of ^211^At-AuNPs and ^211^At-AuNP-PEG-trastuzumab for 24 h.

The obtained results (Figure 5) demonstrated that ^211^At-AuNP-trastuzumab reduced the metabolic activity of SKOV-3 cells with a median lethal dose (LD_50_) of 0.55 MBq mL^−1^. At the same time, ^211^At-AuNPs also reduced cell viability (LD_50_ = 1.3 MBq mL^−1^), although the effect was not as spectacular as in the case with the targeting vector. As has been reported previously, the cytotoxic effect of trastuzumab molecules at the concentrations used in our experiments is negligible [64]. Therefore, we assumed that the cold antibody did not contribute to the cytotoxicity observed with ^211^At-labeled trastuzumab.

Our results confirmed the anticipated highly efficient cell-killing of HER2-expressing breast carcinoma cells by ^211^At-AuNP-trastuzumab at concentrations similar to those described in the literature with other combinations of ^211^At-labeled molecules and tumor cells [64,65,66]. The observed lower cytotoxicity of ^211^At bioconjugates to glioblastoma cells was related to the very high radioresistance of glioblastoma cancer cells. The cytotoxic effect of ^211^At-AuNP-trastuzumab is about seven times greater than that observed with trastuzumab labeled with ^177^Lu, although the half-life of the ^211^At compound is 22 times shorter [67].

Unfortunately, the intravenous injection of our radiobioconjugate for HER2-positive tumors is excluded, due to its relatively large size causing the accumulation of radiobioconjugate in many organs, especially in the liver, lung, and spleen, as described in many papers where nanoparticle-based radiopharmaceuticals were used [68,69].

## 4. Conclusions

Using the discovery of a strong bond formation between the gold surface and astatine atoms, we have successfully synthetized a stable astatinated-trastuzumab bioconjugate. ^211^At-AuNP-trastuzumab was specifically bound, internalized, and distributed to a peri-nuclear location within HER2-positive cancer cells. Trastuzumab-modified ^211^At-AuNPs exhibited higher cytotoxicity than non-targeted ^211^At-AuNPs. These results are encouraging for further development of ^211^At-AuNP-trastuzumab as an innovative α-radiation nanomedicine for local therapy of HER2-positive cancers due to high tumor retention, internalization, and specific tumor cell binding. In comparison to ^177^Lu-AuNP-trastuzumab, the high LET radiation emitted by α-particles gives us the opportunity to increase the effectiveness of targeted nanomedicine brachytherapy for cancer treatment while minimizing healthy tissue toxicity. Nevertheless, due to the short range of α-particles, this form of therapy seems to be most suitable for destroying single micrometastatic cancer cells or small-volume disseminated disease rather than larger tumor masses.

## Figures and Tables

**Figure 1 nanomaterials-09-00632-f001:**
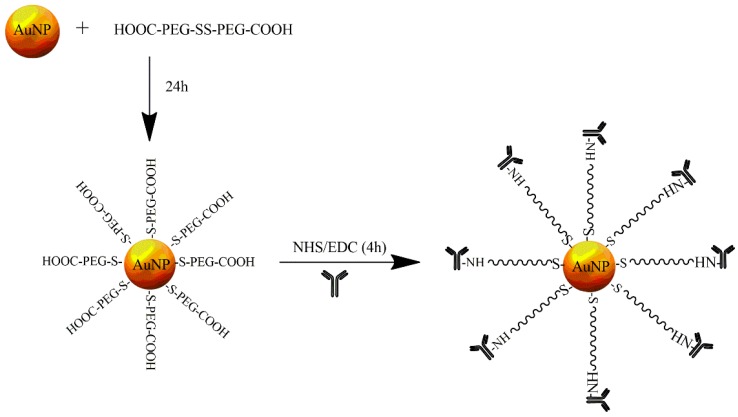
Schematic diagram of trastuzumab to gold nanoparticle (AuNP) surface conjugation.

**Figure 2 nanomaterials-09-00632-f002:**
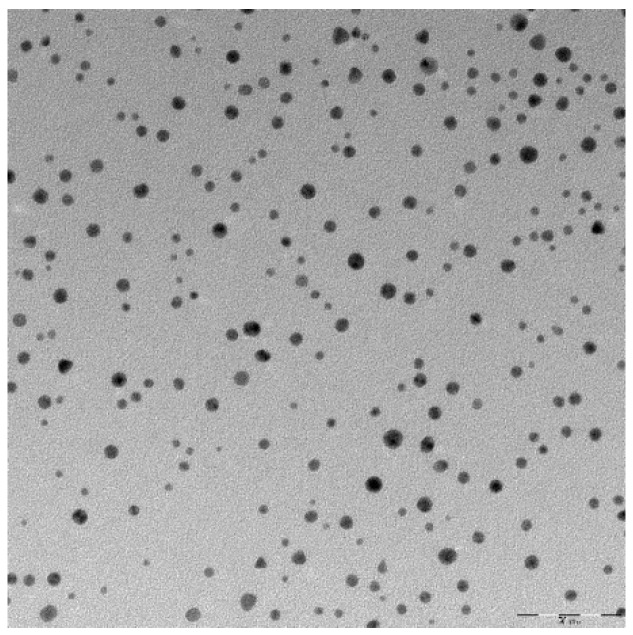
A transmission electron microscopy (TEM) micrograph of the AuNP-S-PEG-trastuzumab bioconjugate.

**Figure 3 nanomaterials-09-00632-f003:**
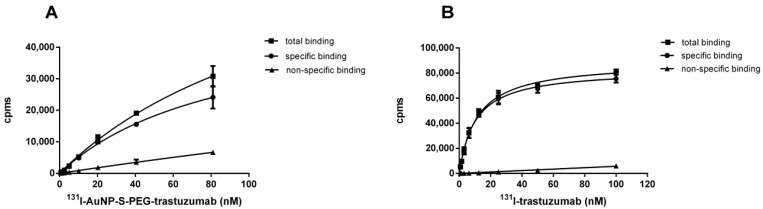
Binding studies of ^131^I-AuNP-S-PEG-trastuzumab (**A**) and ^131^I-trastuzumab (**B**) radiobioconjugates.

**Figure 4 nanomaterials-09-00632-f004:**
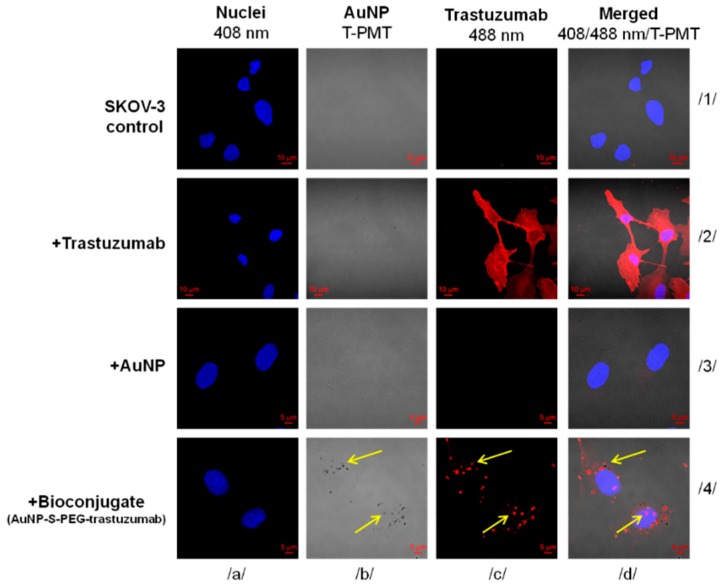
Internalization of free AuNPs and AuNP-S-PEG-trastuzumab bioconjugate by SKOV-3 cells determined by confocal microscopy. SKOV-3 cells that were untreated or treated only with trastuzumab served as positive and negative controls, respectively. Fluorescence signals indicate: (**red**)—subcellular trastuzumab distribution; (**blue**)—nuclei intracellular localization. Au-containing particles (**dark spots**) were visualized with a transmitted light detector (T-PMT). Merged images are presented in column d. Arrows mark the subcellular localization of the AuNP-S-PEG-trastuzumab bioconjugate.

**Figure 5 nanomaterials-09-00632-f005:**
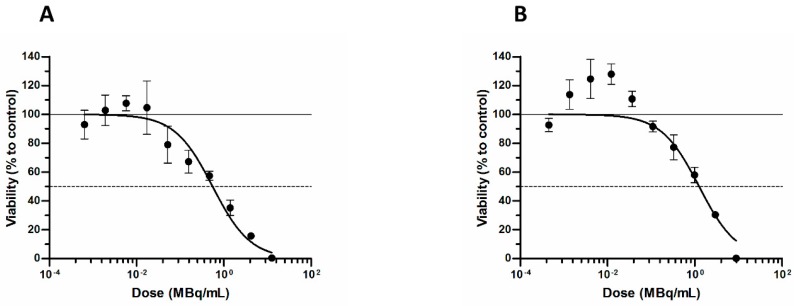
Viability of SKOV-3 cells after treatment with different radioactive doses of: (**A**) ^211^At-AuNP-trastuzumab and (**B**) ^211^At-AuNPs.

**Table 1 nanomaterials-09-00632-t001:** Dynamic Light Scattering (DLS) measurement of hydrodynamic diameters and zeta potentials of 5 nm AuNPs, AuNP-S-PEG-COOH, and AuNP-S-PEG-trastuzumab.

	AuNPs	AuNP-S-PEG-COOH	AuNP-S-PEG-Trastuzumab
Hydrodynamic diameter (nm)	11.7 ± 0.3	16.1 ± 0.5	45.8 ± 3.5
Zeta potential (mV)	−20.2 ± 1.3	−39.6 ± 2.1	−36.9 ± 0.9

**Table 2 nanomaterials-09-00632-t002:** Astatination of AuNPs and AuNP-S-PEG-trastuzumab conjugates with ^211^At.

Nanoparticles of 5 nm	% of Astatination
AuNPs	99.5 ± 0.2
AuNP-S-PEG-trastuzumab	99.6 ± 0.3

**Table 3 nanomaterials-09-00632-t003:** Stability of ^211^At-AuNP-S-PEG-trastuzumab and ^131^I-AuNP-S-PEG-trastuzumab radiobioconjugates in human serum.

Radiobioconjugate	% of Leakage		
	2 h	4 h	24 h
^211^At-AuNP-S-PEG-trastuzumab	0.9 ± 0.2	1.8 ± 0.9	2.4 ± 0.5
^131^I-AuNP-S-PEG-trastuzumab	2.2 ± 0.4	2.6 ± 0.2	2.9 ± 0.3

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
