# Peer review of "Trastuzumab-Modified Gold Nanoparticles Labeled with ^211^At as a Prospective Tool for Local Treatment of HER2-Positive Breast Cancer"

_nanomaterials, 2019, doi:10.3390/nano9040632_

Round 1
Reviewer 1 Report
The work presented by the authors goes beyond the state-of-the-art and is comprehensively presented. This reviewer has a few points for the consideration of the authors:
As the AuNPs are modified with trastuzumab, which provides targeted delivery of the nanoconstruct to the disease site, why do the authors opt for intratumoral injection of the radiolabeled NPs? If the authors further pursue this work, possible disease metastasis will not be treated if the NPs are intratumorally injected.
Was trastuzumab added to the activated with EDG NPs directly after incubation of NPs with EDC and NHS, or after the excess reagents were removed? The authors should clarify this point.
Paragraph 2.6. The authors describe the reaction of 2 x 1011 AuNPs wiwth trastuzumab radiolabeled with 131I- how have the authors determined the number of AuNPs?
Why have the authors used a dialysis membrane to remove the excess of unbound 131I-labeled trastuzumab, which is time-consuming? Could they have used centrifugation with molecula weight cut-off filters, which is a much faster process?
General comment: Upon reading the title, this reviewer expected to see some in vivo results regarding therapeutic efficiency of these novel nanoconstructs on an experimental tumor model. The presented work is of great importance, but is focused on in vitro cell studies. Maybe the authors should consider modifying the title of their manuscript.
Author Response
First of all we acknowledge the Reviewer for their work done.
In the following please find answers for the Reviewer's comments:
As the AuNPs are modified with trastuzumab, which provides targeted delivery of the nanoconstruct to the disease site, why do the authors opt for intratumoral injection of the radiolabeled NPs? If the authors further pursue this work, possible disease metastasis will not be treated if the NPs are intratumorally injected.
The intravenous injection of our radiobioconjugate for HER2 positive tumours is excluded, because of its relatively large size causing the accumulation of radiobioconjugate in many organs, especially in the liver, lung and spleen as described in many papers where nanoparticle-based radiopharmaceuticals were used. These limitations have stimulated the development of strategies for local drug application. According to our concept, 211At-AuNP-PEG-trastuzumab will be injected into the cavity after surgical resection of the tumor and should destroy the remaining clusters or single cancer cells. The relevant fragment is added to the conclusions.
Was trastuzumab added to the activated with EDG NPs directly after incubation of NPs with EDC and NHS, or after the excess reagents were removed? The authors should clarify this point.
Trastuzumab was added directly to the activated with EDC/NHS gold nanoparticles. Further, the excess of EDC/NHS, free trastuzumab and free PEG was removed by dialysis using 30kDa MW cutoff membrane.
Paragraph 2.6. The authors describe the reaction of 2 x 1011 AuNPs with trastuzumab radiolabeled with 131I- how have the authors determined the number of AuNPs?
Firstly we assumed that all 5 nm AuNPs are spherical and we calculated the volume. Next we determined the mass of each nanoparticles. Knowing the mass of gold used for synthesis we determined the number of obtained nanoparticles. Of course the amounts of AuNPs is only estimation due to some assumptions.
Why have the authors used a dialysis membrane to remove the excess of unbound 131I-labeled trastuzumab, which is time-consuming? Could they have used centrifugation with molecula weight cut-off filters, which is a much faster process?
We agree that dialysis is a time consuming process but we had a problem with the centrifugation of 5 nm AuNPs using Vivaspins. The AuNPs physically adsorbed on the cut-off filters. It was very difficult to remove them from the membrane. In such situation we decided to use a dialysis.
General comment: Upon reading the title, this reviewer expected to see some in vivo results regarding therapeutic efficiency of these novel nanoconstructs on an experimental tumor model. The presented work is of great importance, but is focused on in vitro cell studies. Maybe the authors should consider modifying the title of their manuscript.
Thank you for valuable remark. We modified the title for “Trastuzumab modified gold nanoparticles labeled with 211At as a perspective tool for local treatment of HER2-positive breast cancer”

Reviewer 2 Report
Review for Nanomaterials-469095
General comments
The authors present the application of 211-At-AuNP-PEG-trastuzumab radio-bio-conjugate as a novel therapeutic agent for the treatment of un-resectable HER2-positive breast cancer. The methods are presented in a thorough and systematic manner while the reported results are very encouraging. This is a significant work with new and useful findings for future investigators.
However, the authors should explain and justify with appropriate references the choice for the various methods employed to realize each step presented in the methods section and whenever necessary support their statements with appropriate references. The syntax/grammar must also be improved to ensure sufficient presentation quality for a scientific manuscript. In the following section, multiple issues are raised and potential revisions/editing is suggested to improve the quality and potential of the manuscript under consideration. The authors should address point-by-point both the general and specific comments in the manuscript and response to reviewers before further considerations can be issued.
Specific Comments
Abstarct, Line 25, “TEM and DLS techniques”: Throughout the manuscript, the authors should spell out abbreviations at the first time they are used. The same should hold for the abstract. The authors should thus explain here the abbreviated terms "TEM" and "DLS".
Abstract, Line 33, “for 211-AT-AuNP-PEG-trastuzumab”: The authors do not clearly indicate in the abstract if the modified gold nanoparticles with PEG chain and trastuzumab performed better than conventional gold nanoparticles. This is a major finding and, as such, it should be fairly represented in the abstract.
Page 2, Line 46, “125-I, 106-Ru … and 90-Y immobilized…”: References for the use of each of those radioactive seeds should be included for the justification of this statement.
Page 2, Lines 49-51, “The limited affinity … from the tumor site”: The authors should support this statement with an adequate number of references.
Page 2, Lines 53-56, “With the rapid development… surgical implantation”: Both of these statement should be accompanied by relevant references supporting them.
Page 2, Line 64, “suggested use of silica nanoparticles…”: The following revision should be issued: "suggested the use of...".
Page 2, Line 77-79, “They also found that targeting HER2 … with non-functionalized AuNP-177Lu.”: This sentence needs to be revised for better clarity and readability. The qualitative characterization of “dramatically greater” needs to be quantified with % indices. What did the author mean by the adjective “functionalized” for the AUNPs?
Page 2, Line 84, 85: The range of alpha particles… and its close neighbors”: The authors should provide adequate references to justify this statement.
Page 2, Line 94, “remain limited to isolated by products”: This syntax does not make sense. The authors should revise it such that readability and clarity is enhanced.
Page 3, Line 98-99, “without problems caused by… emitting alpha particles”: The authors should explain the type of problems expected by the relatively long-lived daughter particles and provide relevant references.
Page 3, Line 111, “to extensions of a new method”: The following revision should be issued: "to introduce a new method for iodination..."
Page 4, Line 154-155, “Due to better availability of 131-I… presented experiments”: The authors should specify what properties similar between the two isotopes made I-131 a good substitute for 221-AT and provide references that justify this statement.
Page 4, Line 158, “2.3 Characterizations”: The title "characterizations" is rather vague and not quite reflect the content of this paragraph. A more representative title should be used. In addition, all the multiple steps of the various processes described in this and subsequent 2.x paragraphs should be accompanied by an adequate number of references of previous publications employing similar processes for similar purposes.
Page 4, Line 160, “by transmission electron microscopy”: References should be provided introducing the basic principles of TEM and its applications relevant to this study.
Page 4, Line 161, “by dynamic light scattering”: References should be provided introducing the basic principles of DLS and being used to determine the hydrodynamic diameter and zeta potential, as suggested in this study
Page 4, line 167, “… the electrolyte was applied”: The verb needs to be places earlier in this statement, e.g. by avoiding the passive form, to enhance readability.
Page 4, Line 174, “T-PMT”: All used abbreviations should be spelled out explicitly in the manuscript when first used.
Page 4, Line 177-179, “Briefly, to boiling … was added”: The verb of this sentence should be placed earlier in the sentence. The following revision is suggested: "More specifically, a mixture of... was added to a boiling aqueous solution of..."
Page 4, Line 180, “a further 15min”: The following revision should be issued to enhance clarity and readability: "was stirred for an additional 15min period and was finally cooled down ..."
Page 4, Line 187, “After a further 4h”: The phrase "further 15min or "further 4h" etc used in this manuscript is not quite appropriate and should be replaced as suggested in a previous related comment.
Page 5, Line 195-196, “using the Iodogen method”: The Iodogen method should be briefly explained in the manuscript and accompanied by an appropriate reference.
Page 5, Lines 211-214, “To determine the labeling efficiency… at the bottom of the TLC strip”: References with previous application of the TLC method with MeOH should be included
Page 5, Line 237, “bioconiugate”: The authors should correct the typo error.
Page 6, Line 251-252, “was estimated using Scatchard analysis method”: References to the Scatchard analysis method should be included
Page 6, Line 261-262, “Cell viability was expressed… in medium only”: This statement needs to be revised to make clear sense. It is not quite clear, what the authors meant by "... expressed as a percentage by normalization..."
Page 6, Line 281-282, “The promising results… with trastuzumab… led us to us…”: The authors should cite the publication referring to those result in this statement.
Page 6, Line 283, “Our group discovered…”: The authors should avoid very ling paragraphs throughout the manuscript. It is suggested that that a new paragraph begins here.
Page 8, Line 343-344, “This is expected because… of the AuNP surface”: The authors should justify this statement with appropriate references.
Page 11, Line 428, “with trastuzumab-labeled with the beta minus emitter”: This expression needs to be rephrased for clarity. The use of the term “with twice in this statement is confusing and should be revised.
Page 11, Line 429, “the latter compound”: Do the authors refer to Lu-177 half-life? A shorter half-life usually indicates lower cytotoxic effects. Why the opposite is suggested here? The authors should explain more clearly in the manuscript why they use the term "although" if 177-Lu is associated with much shorter half-life.
Author Response
First of all we acknowledge the Reviewer for their work done.
In the following please find answers for the Reviewer's comments:
General comments
The authors present the application of 211-At-AuNP-PEG-trastuzumab radio-bio-conjugate as a novel therapeutic agent for the treatment of un-resectable HER2-positive breast cancer. The methods are presented in a thorough and systematic manner while the reported results are very encouraging. This is a significant work with new and useful findings for future investigators.
However, the authors should explain and justify with appropriate references the choice for the various methods employed to realize each step presented in the methods section and whenever necessary support their statements with appropriate references. The syntax/grammar must also be improved to ensure sufficient presentation quality for a scientific manuscript. In the following section, multiple issues are raised and potential revisions/editing is suggested to improve the quality and potential of the manuscript under consideration. The authors should address point-by-point both the general and specific comments in the manuscript and response to reviewers before further considerations can be issued.
Specific Comments
Abstarct, Line 25, “TEM and DLS techniques”: Throughout the manuscript, the authors should spell out abbreviations at the first time they are used. The same should hold for the abstract. The authors should thus explain here the abbreviated terms "TEM" and "DLS".
Thank you for the valuable remark. We explained the abbreviated terms “TEM” and “DLS”.
Abstract, Line 33, “for 211-AT-AuNP-PEG-trastuzumab”: The authors do not clearly indicate in the abstract if the modified gold nanoparticles with PEG chain and trastuzumab performed better than conventional gold nanoparticles. This is a major finding and, as such, it should be fairly represented in the abstract.
The abstract is changed.
Page 2, Line 46, “125-I, 106-Ru … and 90-Y immobilized…”: References for the use of each of those radioactive seeds should be included for the justification of this statement.
We added references to the text. Thank you.
Page 2, Lines 49-51, “The limited affinity … from the tumor site”: The authors should support this statement with an adequate number of references.
We added reference to the text. Thank you.
Page 2, Lines 53-56, “With the rapid development… surgical implantation”: Both of these statement should be accompanied by relevant references supporting them.
We added references to the text. Thank you.
Page 2, Line 64, “suggested use of silica nanoparticles…”: The following revision should be issued: "suggested the use of...".
We modified this sentence in the text.
Page 2, Line 77-79, “They also found that targeting HER2 … with non-functionalized AuNP-177Lu.”: This sentence needs to be revised for better clarity and readability. The qualitative characterization of “dramatically greater” needs to be quantified with % indices. What did the author mean by the adjective “functionalized” for the AUNPs?
Thank you for valuable remark. We modified the sentence for: “They also found that targeting HER2 with trastuzumab facilitated the binding and internalization in HER2 positive tumor cells in comparison to nontargeted AuNP-177Lu”
“Functionalized AuNPs” means AuNPs conjugated with trastuzumab.
Page 2, Line 84, 85: The range of alpha particles… and its close neighbors”: The authors should provide adequate references to justify this statement.
We added the reference.
Page 2, Line 94, “remain limited to isolated by products”: This syntax does not make sense. The authors should revise it such that readability and clarity is enhanced.
We revised the sentence for better clarity and readability. Thank you.
Page 3, Line 98-99, “without problems caused by… emitting alpha particles”: The authors should explain the type of problems expected by the relatively long-lived daughter particles and provide relevant references.
We revised the text. Reference is added.
Page 3, Line 111, “to extensions of a new method”: The following revision should be issued: "to introduce a new method for iodination..."
We corrected, thank you.
Page 4, Line 154-155, “Due to better availability of 131-I… presented experiments”: The authors should specify what properties similar between the two isotopes made I-131 a good substitute for 221-AT and provide references that justify this statement.
Astatine and iodine are neighboring elements in the halogen group and therefore they have a number of similar properties, such as stability of oxidation state -1. The appropriate text is added.
Page 4, Line 158, “2.3 Characterizations”: The title "characterizations" is rather vague and not quite reflect the content of this paragraph. A more representative title should be used. In addition, all the multiple steps of the various processes described in this and subsequent 2.x paragraphs should be accompanied by an adequate number of references of previous publications employing similar processes for similar purposes.
We corrected the title “characterizations” for “characterization techniques for nanoparticles”.
Page 4, Line 160, “by transmission electron microscopy”: References should be provided introducing the basic principles of TEM and its applications relevant to this study.
We added reference to the text. Thank you.
Page 4, Line 161, “by dynamic light scattering”: References should be provided introducing the basic principles of DLS and being used to determine the hydrodynamic diameter and zeta potential, as suggested in this study.
We added reference to the text. Thank you.
Page 4, line 167, “… the electrolyte was applied”: The verb needs to be places earlier in this statement, e.g. by avoiding the passive form, to enhance readability.
We corrected the sentence, thank you.
Page 4, Line 174, “T-PMT”: All used abbreviations should be spelled out explicitly in the manuscript when first used.
We explained the abbreviated term ”T-PMT” in the text.
Page 4, Line 177-179, “Briefly, to boiling … was added”: The verb of this sentence should be placed earlier in the sentence. The following revision is suggested: "More specifically, a mixture of... was added to a boiling aqueous solution of..."
We corrected the sentence regarding to your suggestion.
Page 4, Line 180, “a further 15min”: The following revision should be issued to enhance clarity and readability: "was stirred for an additional 15min period and was finally cooled down ..."
We corrected the sentence regarding to your suggestion.
Page 4, Line 187, “After a further 4h”: The phrase "further 15min or "further 4h" etc used in this manuscript is not quite appropriate and should be replaced as suggested in a previous related comment.
We corrected the sentence regarding to your suggestion.
Page 5, Line 195-196, “using the Iodogen method”: The Iodogen method should be briefly explained in the manuscript and accompanied by an appropriate reference.
We described briefly the Iodogen method in the manuscript and added the reference.
Page 5, Lines 211-214, “To determine the labeling efficiency… at the bottom of the TLC strip”: References with previous application of the TLC method with MeOH should be included
We included the reference.
Page 5, Line 237, “bioconiugate”: The authors should correct the typo error.
We corrected, thank you.
Page 6, Line 251-252, “was estimated using Scatchard analysis method”: References to the Scatchard analysis method should be included
We included the reference, thank you.
Page 6, Line 261-262, “Cell viability was expressed… in medium only”: This statement needs to be revised to make clear sense. It is not quite clear, what the authors meant by "... expressed as a percentage by normalization..."
We revised the sentence in the manuscript.
Page 6, Line 281-282, “The promising results… with trastuzumab… led us to us…”: The authors should cite the publication referring to those result in this statement.
We included the publication referring this statement in the manuscript.
Page 6, Line 283, “Our group discovered…”: The authors should avoid very ling paragraphs throughout the manuscript. It is suggested that that a new paragraph begins here.
We changed the paragraph according to your suggestion.
Page 8, Line 343-344, “This is expected because… of the AuNP surface”: The authors should justify this statement with appropriate references.
This phenomenon resulted from the fact that we chose the ratio of trastuzumab modified molecules to gold nanoparticles in such a way that the half of the gold surface was free for attaching 211At. The text is corrected.
Page 11, Line 428, “with trastuzumab-labeled with the beta minus emitter”: This expression needs to be rephrased for clarity. The use of the term “with twice in this statement is confusing and should be revised.
We rephrased this expression. Thank you.
Page 11, Line 429, “the latter compound”: Do the authors refer to Lu-177 half-life? A shorter half-life usually indicates lower cytotoxic effects. Why the opposite is suggested here? The authors should explain more clearly in the manuscript why they use the term "although" if 177-Lu is associated with much shorter half-life.
It was our mistake. Instead of “the latter” should be written “the 211At”. 211At has the shorter half-life in comparison to 177Lu but due to high LET (as an alpha emitter) exhibits better cytotoxicity. We corrected the sentence “ The cytotoxic effect of 211At-AuNP-trastuzumab is about seven times greater than observed with trastuzumab labeled with 177Lu, although the half-life of the 211At compound is 22 times shorter”

Reviewer 3 Report
Nice work.
Author Response
Thank you very much for a nice and positive review.

Round 2
Reviewer 1 Report
All the reviewer's comments provided by this reviewer have been satisfactorily addressed.
Author Response
Thank you very much for a nice review.
Reviewer 2 Report
General Comments
The authors have clarified the major comments of the previous review cycle and addressed most of the issues in their response to the authors. However, it is important that the provided clarifications/explanations are also included in the manuscript to ensure clarity and readability of the manuscript and all the methodological details assumed in the presented work. The authors are therefore strongly advised to provide all detailed explanation in the manuscript text as well. This is necessary before further consideration can be made. More details can be found in the specific comments section below
Specific comments
The authors have responded to the following review comment: “Was trastuzumab added to the activated with EDG NPs directly after incubation of NPs with EDC and NHS, or after the excess reagents were removed? The authors should clarify this point.” However, their response does not appear in the manuscript text. They should clearly designate the point in the text manuscript where this explanation is provided.
The authors have responded to the following review comment: “Paragraph 2.6. The authors describe the reaction of 2 x 1011 AuNPs with trastuzumab radiolabeled with 131I- how have the authors determined the number of AuNPs?”. However, similarly to the previous comments, their provided elaboration does not seem to be included in the new revised manuscript. The authors should revise the manuscript accordingly and designate the revised section to the reviewers.
The authors have responded to the following review comment: “Why have the authors used a dialysis membrane to remove the excess of unbound 131I-labeled trastuzumab, which is time-consuming? Could they have used centrifugation with molecular weight cut-off filters, which is a much faster process?”. However, again they should ensure their explanation is clearly presented in their revised manuscript as well.
In Conclusions, lines 448-451, the added statement is mostly suitable for the Discussion section rather than the Conclusion section. The authors are advised to move this statement in the discussion section. Moreover, this statement should be justified with adequate number of references.
Author Response
We would like to thank Reviewer very much for all the corrections and comments. We corrected the manuscript according to suggestions and we believe that after that improvements the paper is much more clear and better to read.
General Comments
The authors have clarified the major comments of the previous review cycle and addressed most of the issues in their response to the authors. However, it is important that the provided clarifications/explanations are also included in the manuscript to ensure clarity and readability of the manuscript and all the methodological details assumed in the presented work. The authors are therefore strongly advised to provide all detailed explanation in the manuscript text as well. This is necessary before further consideration can be made. More details can be found in the specific comments section below.
Specific comments
The authors have responded to the following review comment: “Was trastuzumab added to the activated with EDG NPs directly after incubation of NPs with EDC and NHS, or after the excess reagents were removed? The authors should clarify this point.” However, their response does not appear in the manuscript text. They should clearly designate the point in the text manuscript where this explanation is provided.
We added the explanation to the text (lines 192-193).
The authors have responded to the following review comment: “Paragraph 2.6. The authors describe the reaction of 2 x 1011 AuNPs with trastuzumab radiolabeled with 131I- how have the authors determined the number of AuNPs?”. However, similarly to the previous comments, their provided elaboration does not seem to be included in the new revised manuscript. The authors should revise the manuscript accordingly and designate the revised section to the reviewers.
We included the sentence describing the determination of the number of AuNPs to the text (lines 183-185).
The authors have responded to the following review comment: “Why have the authors used a dialysis membrane to remove the excess of unbound 131I-labeled trastuzumab, which is time-consuming? Could they have used centrifugation with molecular weight cut-off filters, which is a much faster process?”. However, again they should ensure their explanation is clearly presented in their revised manuscript as well.
We added appropriate explanation to the text (line 206-207). Thank you.
In Conclusions, lines 448-451, the added statement is mostly suitable for the Discussion section rather than the Conclusion section. The authors are advised to move this statement in the discussion section. Moreover, this statement should be justified with adequate number of references.
We moved the sentence to the discussion part and added the references (lines 441-444).

Round 3
Reviewer 2 Report
Review Report for Nanomaterials-469095 v3
General Comments
The authors have addressed the minor comments of the previous review cycle in their response to the reviewers. However, the revised sections in the manuscript still need a certain degree of editing of syntax and clarifications to ensure an accurate and clear message is communicated to the reader.
In the specific comment section below, the authors are suggested additional minor revisions that are important for the correct understanding and assimilation of the methods and conclusions of the presented study.
Specific Comments
Abstract, Lines 34-35, "as a therapeutic agent": Based on the scope of this study, the therapeutic efficacy in the clinic has not yet been adequately demonstrated and thus the authors should avoid definitive statements in the abstract. The following revision is therefore suggested: "a potential therapeutic agent in..."
Line 185, "numbers of AuNPs..": The following syntax correction should be applied: "the numbers of..."
Line 192, "Directly, ...": The use of the term "Directly" is not appropriate here. It is not quite clear what is the purpose of the use of this term at this point. Thus, the authors are advised to remove this term from the beginning of their sentence to avoid creating confusion and to enhance readability.
Line 205-206, "Due to the impossibility... was applied": The following syntax revision is strongly suggested to the authors to enhance readability and clarity of their message to the reader: "As the centrifugation of 5nm size AuNPs is not possible, due to their small size, we applied the dialysis method with the use of a membrane of 30 kDa instead of Vivaspins"
Author Response
Thank you for the review. In the following please find answers for the Reviewer's comments:
General Comments
The authors have addressed the minor comments of the previous review cycle in their response to the reviewers. However, the revised sections in the manuscript still need a certain degree of editing of syntax and clarifications to ensure an accurate and clear message is communicated to the reader.
In the specific comment section below, the authors are suggested additional minor revisions that are important for the correct understanding and assimilation of the methods and conclusions of the presented study.
Specific Comments
Abstract, Lines 34-35, "as a therapeutic agent": Based on the scope of this study, the therapeutic efficacy in the clinic has not yet been adequately demonstrated and thus the authors should avoid definitive statements in the abstract. The following revision is therefore suggested: "a potential therapeutic agent in..."
We corrected the sentence.
Line 185, "numbers of AuNPs..": The following syntax correction should be applied: "the numbers of..."
We corrected.
Line 192, "Directly, ...": The use of the term "Directly" is not appropriate here. It is not quite clear what is the purpose of the use of this term at this point. Thus, the authors are advised to remove this term from the beginning of their sentence to avoid creating confusion and to enhance readability.
We removed the word “directly” from the text, thank you.
Line 205-206, "Due to the impossibility... was applied": The following syntax revision is strongly suggested to the authors to enhance readability and clarity of their message to the reader: "As the centrifugation of 5nm size AuNPs is not possible, due to their small size, we applied the dialysis method with the use of a membrane of 30 kDa instead of Vivaspins"
Thank you for valuable remark. We corrected the sentence regarding to your suggestion.
